# A Review of the Systemic Treatment of Stevens–Johnson Syndrome and Toxic Epidermal Necrolysis

**DOI:** 10.3390/biomedicines10092105

**Published:** 2022-08-28

**Authors:** Hua-Ching Chang, Tsung-Jen Wang, Ming-Hsiu Lin, Ting-Jui Chen

**Affiliations:** 1Department of Dermatology, School of Medicine, College of Medicine, Taipei Medical University Hospital and Taipei Medical University, Taipei 11031, Taiwan; 2Research Center of Big Data and Meta-Analysis, Wan Fang Hospital, Taipei Medical University, Taipei 11031, Taiwan; 3Department of Ophthalmology, School of Medicine, College of Medicine, Taipei Medical University, Taipei 11031, Taiwan; 4Department of Ophthalmology, Taipei Medical University Hospital, Taipei 11031, Taiwan; 5Department of Dermatology, Taoyuan General Hospital, Ministry of Health and Welfare, Taoyuan 33004, Taiwan

**Keywords:** Stevens–Johnson syndrome, toxic epidermal necrolysis, meta-analysis

## Abstract

Stevens–Johnson syndrome (SJS) and toxic epidermal necrolysis (TEN) are uncommon but life-threatening diseases mostly caused by drugs. Although various systemic immunomodulating agents have been used, their therapeutic efficacy has been inconsistent. This study aimed to provide an evidence-based review of systemic immunomodulating treatments for SJS/TEN. We reviewed 13 systematic review and meta-analysis articles published in the last 10 years. The use of systemic corticosteroids and IVIg is still controversial. An increasing number of studies have suggested the effectiveness of cyclosporine and biologic anti-TNF-α in recent years. There were also some promising results of combination treatments. Further large-scale randomized controlled trials are required to provide more definitive evidence of the effectiveness of these treatments. The pathogenesis of SJS/TEN has been elucidated in recent years and advances in the understanding of SJS/TEN may inspire the discovery of potential therapeutic targets.

## 1. Introduction

Stevens–Johnson syndrome (SJS) and toxic epidermal necrolysis (TEN) are severe cutaneous adverse reactions mostly caused by medications. The two conditions are considered as the same disease with different spectrums of severity. SJS is defined as skin detachment of less than 10% of body surface area, while TEN involves skin detachment greater than 30%. Overlapping SJS–TEN is defined as 10–30% skin detachment [1]. Based on the National Health Insurance database in Korea, the incidence of SJS in adults ranges from 3.96 to 5.3/1,000,000 and that of TEN ranges from 0.4 to 1.45/1,000,000 population [2]. Extensive skin detachment in SJS/TEN may cause significant morbidity and mortality. The reported mortality rate in patients with SJS and TEN is 4.8% and 14.8%, respectively [3]. SJS/TEN is considered as one of the few dermatological emergencies, and early recognition and appropriate management may save lives [1].

SJS/TEN is characterized by painful blisters, purpuric macules, and atypical target lesions with both skin and mucosal involvement. Lesions typically begin to appear 4–28 days after initiation of the culprit drug. The skin rash is often preceded by malaise, fever, and upper respiratory tract (flu-like) symptoms. Almost all patients with SJS/TEN have mucosal involvement in eyes, mouth, and genitalia. In addition to the skin and mucosal involvement, other organ systems such as cardiovascular, pulmonary, gastrointestinal, and urinary tract system may also be affected. Multiple organ involvement may cause complications and sequalae. Skin infection, pneumonia, hepatitis and sepsis are frequently reported complications of SJS/TEN, which may cause mortality [4].

In addition to the high mortality, sequalae are also commonly found in the recovery phase of SJS/TEN. The most common sequalae are skin and mucosal problems. Hoffman et al. reported the prevalence of physical sequalae relevant to SJS/TEN, including cutaneous problems (84.3%), ocular problems (59.5%), and oral mucosal problems (50.8%) [5]. Skin hyperpigmentation, scarring, hair loss, and nail dystrophy are also common. Ocular complications include dry eyes, foreign body sensation, chronic conjunctivitis, trichiasis, and even blindness. Psychological problems including anxiety, depression, and post-traumatic stress disorder have also been reported in these patients. According to a study, the physical and psychological sequelae of SJS/TEN caused lack of ability to work in 28.2% patients, and 68.1% and 30.0% patients were fearful or avoided taking medications, respectively [5].

Although SJS/TEN is a distressing disease, there is no standard treatment for SJS/TEN. The rarity of SJS/TEN is the barrier that makes it difficult to conduct high-quality double-blind controlled studies to elucidate the efficacy of medications. Most of the contemporary literature comprises of retrospective studies with no clear consensus on the effectiveness of different therapies. In this review, we aimed to provide evidence-based review of systemic immunomodulatory treatment of SJS/TEN.

## 2. Pathogenesis

Prominent keratinocyte apoptosis with epidermal necrosis and dermo-epidermal separation are the hallmark histopathological features of skin lesions in SJS/TEN. The death of keratinocytes is believed to be induced by CD8 cytotoxic T cells and natural killer cells through an interaction with the human leukocyte antigen (HLA) and drug antigens [6]. An increasing body of evidence has shown an association of genetic factors with higher incidence of SJS/TEN. In 2004, Chung et al. reported a strong association between HLA-B*15:02 and carbamazepine (CBZ)-induced SJS/TEN in a Han Chinese population. Similar results were reported in Asian populations of Thailand and Malaysia. However, no similar association was found in Japanese, Korean, or European populations [7].

Three pathways are known to contribute to the keratinocyte apoptosis observed in SJS/TEN: Fas–Fas ligand (FasL) interaction, perforin/granzyme B, and granulysin. Fas–Fas ligand-induced apoptosis of keratinocytes was originally hypothesized by Viard et al. [8]. Under normal conditions, Fas is present on the surface of keratinocytes. Viard et al. demonstrated the presence of FasL on the surface of keratinocytes along with high serum levels of soluble FasL (sFasL) in TEN patients, but not in patients with maculopapular drug eruption or healthy individuals [8].

In 1997, Inachi et al. demonstrated dermal perforin-positive cells infiltration in SJS skin lesions and suggested the involvement of perforin in the pathogenesis of keratinocyte apoptosis in SJS [9]. Perforin, a pore-making protein released from natural killer cells and cytotoxic T lymphocytes, kills target cells by forming polymers and tubular structures on the cell membrane [9]. Nassif et al. demonstrated that lymphocytes in the blister fluid of TEN patients are cytotoxic in the presence of a causative drug. This cytotoxicity can be blocked by the perforin/granzyme pathway inhibitor. These findings suggest that perforin/granzyme B also play an important role in inducing keratinocyte death in SJS/TEN [10,11].

In 2004, Chung et al. identified granulysin as another key mediator of SJS/TEN. Granulysin is released from CD8+ cytotoxic T cells and natural killer cells. A study demonstrated high levels of granulysin in the blister fluid of SJS/TEN patients and the granulysin levels in blister fluid showed a positive correlation with the clinical severity [12]. Further study suggested that serum granulysin level may also serve as an early diagnostic marker of SJS/TEN [13].

Various cytokines are reportedly involved in SJS/TEN. In 2004, Amal et al. demonstrated elevated levels of interferon gamma (IFN-γ), soluble tumor necrosis factor alpha (TNF-α), and sFasL in the blister fluid of 13 TEN patients [14]. In 2017, Su et al. detected upregulated serum levels of IL-6, IL-8, IL-15, TNF-α, and granulysin in 33 SJS/TEN patients. Consistently, analysis of 155 samples (77 samples from the Taiwan Severe Cutaneous Adverse Reaction Consortium, and 78 from the International Registry of Severe Cutaneous Adverse Reactions to drugs) revealed a significant correlation of IL-15 levels with disease severity and prognosis. IL-15 plays a central role in the maintenance of cytotoxic T cell responses and NK-cell functions. IL-15 directly induces the production of TNF-α and is involved in the elevation of several cytokines/chemokines, including IL-1, IL-6, and IL-8 [15].

The pathophysiology of SJS/TEN is now believed to involve immune-mediated reactions from both innate and adaptive immune systems. Therefore, use of various systemic immunomodulating agents including steroids, intravenous immunoglobulin (IVIg), cyclosporine, and anti-TNF-α has been investigated for stopping the progression of epidermal necrosis. However, many of the reports pertaining to these therapies were uncontrolled studies and there is no consistent evidence of the therapeutic effectiveness.

## 3. Literature Review for Systemic Treatment for SJS/TEN

Owing to the recent advances in the understanding of pathogenesis, supportive care, and wound care, studies published in recent years may provide a more accurate comparison of treatment and supportive care. Therefore, we identified studies published in PubMed from 1 January 2012 to February 2022. All articles included in the current review were clinical studies published in English. The search parameters included the terms “toxic epidermal necrolysis” or “Stevens-Johnsons syndrome” combined with “systematic review” or “meta-analysis”. A total of 13 relevant studies were included and summarized in Table 1 [16,17,18,19,20,21,22,23,24,25,26,27,28].

## 4. Management

Due to the high morbidity and mortality of SJS/TEN, multidisciplinary care in a specialized burn unit is recommended for these patients. UK guidelines suggest transferring patients to a Burn Centre for patients with TEN and evidence of the following manifestations: clinical deterioration, extension of epidermal detachment, sub-epidermal pus, local sepsis, wound conversion and/or delayed healing [29].

According to the UK guidelines for the management of SJS/TEN, withdrawal of the culprit drug and multidisciplinary supportive care should be prioritized over systemic treatment because of the paucity of evidence of the treatment efficacy [29]. However, Japanese guidelines for SJS/TEN recommend early systemic corticosteroids, either alone or in combination with cyclosporin as the first-line treatment [30].

### 4.1. Culprit Drugs Identification and Withdrawal

Identification and withdrawal of the culprit drug is the most crucial part of the management. Culprit drugs have been reportedly identified in 85% of cases of SJS/TEN [31]. In some cases, identification of the culprit drug is difficult, especially in patients taking multiple drugs concurrently. The ALDEN (ALgorithm of Drug causality for Epidermal Necrolysis) algorithm is generally used for assessment of drug causality retrospectively but not in the acute phase [31]. Pharmacovigilance data play an important role in identifying drugs that have a very strong association with SJS/TEN [32,33].

Several tests have been used for identifying the culprit drug. Oral provocation test with the culprit drug is generally considered to be the “gold standard” for most drug reactions, but it is not recommended for severe and hazardous reactions such as SJS/TEN. Patch testing has been used to identify the culprit drug of cutaneous adverse reactions such as acute generalized exanthematous pustulosis (AGEP), maculopapular exanthema, or drug rashes with eosinophilia and systemic symptoms (DRESS). However, there are no standard preparations of testing agent. Wolkenstein et al. reported 50% positive rates for AGEP, but only two patients among the 22 SJS/TEN cases had a positive test [34]. A multi-center study reported similar results with the culprit drug identified in 64% (46/72) of patients with DRESS, 58% (26/45) of patients with AGEP, and only 24% (4/17) of those with SJS/TEN [35]. In contrast, Lin et al. reported positive patch test in 62.5% (10/16) of patients with carbamazepine-induced SJS/TEN in a single-center study and there was no recurrence of hypersensitivity reaction during or after patch testing. Cross-sensitivity to structure-related aromatic anti-epileptic drugs such as oxcarbazepine, phenytoin, and lamotrigine was observed in patients with carbamazepine-induced SJS/TEN. The authors suggested that those aromatic anti-epileptics should be avoided in these cases [36]. In conclusion, patch testing shows highly variable sensitivity and specificity for different drugs.

Culprit drugs may also potentially be identified using in vitro assays, including lymphocyte transformation tests (LTT) and drug-induced lymphocyte cytokine production (cytokine assays). The LTT measures the proliferation of T cells in response to a drug in vitro and has been reported positive in 21–56% of patients with SJS/TEN [37,38,39,40]. Recently, researchers have modified the LTT in order to improve the specificity by adding IL-2, IL-7/IL-15, professional antigen presenting cells or by removal of regulatory T cells (CD3+ CD25+) [41]. Cytokine assays measure levels of cytokines or other mediators produced by lymphocytes secondary to a drug stimulation. Many cytokines such as IFN- γ, IL-2, IL-4, and IL-5 are expressed and released during the delayed-type drug hypersensitivity test. In recent studies, IFN-γ assays and IL-4 assay were found to identify the culprit drug in 78% and 50% of cases of SJS/TEN, respectively [39]. In another study, the culprit drug was identified in 55% of cases with IFN-γ assays, in 43% of cases with IL-5 assay, and in 38% of cases with IL-2 assay [40]. The study also suggested that combining different assays may be a more feasible approach to identify the culprit drugs in patients with SJS/TEN. Currently, LTT and cytokine assays are not routinely used in clinical settings for identification of the culprit drug in patients with SJS/TEN.

### 4.2. Severity-of-Illness Score for TEN (SCORTEN)

The well-known severity-of-illness score for TEN (SCORTEN) is widely used to evaluate and predict mortality due to SJS/TEN [42]. SCORTEN consists of seven independent risk factors, including age > 40 years, malignancy, tachycardia > 120 beats/minute, skin detachment > 10%, serum urea > 10 mmol/L, glucose > 14 mmol/L, and serum bicarbonate < 20 mmol/L. SCORTEN should be assessed within the first 24 h and on day 3 after admission.

### 4.3. Supportive Therapy

The supportive care for SJS/TEN patients is similar to the management of a severe burn patient. It encompasses protecting and restoring the barrier function of the skin, maintaining fluid balance, protecting the airway, and treating infection. Fluid and electrolyte monitoring and replacement are essential. Nutritional support is also essential due to the high catabolic state. Furthermore, thermoregulation and adequate analgesia are usually needed [43,44]. There are no clinical guidelines for the skin care of patients with SJS/TEN. Debridement of necrotic epidermis was recommended in the past but considered unnecessary in recent years. Detached epidermis is considered a natural biologic dressing which hastens re-epithelialization [45].

Multisystem involvement also requires early initiation of multidisciplinary care involving experts from the departments of gynecology, urology, colorectal, ear, nose, and throat (ENT), and ophthalmology to help prevent the sequelae of SJS/TEN. Consulting the ophthalmologist is essential because most patients have ocular involvement. Treatment with aggressive lubrication, topical antibiotics, topical corticosteroids, and lysis of adhesions for eyes is necessary. Recently, amniotic membrane transplant has been shown to be effective in preserving visual acuity and an intact ocular surface [43]. Since SJS/TEN may cause psychological impact, appropriate information and emotional support for the patients and their families are important [5].

To summarize, the most important aspects of SJS/TEN management are early diagnosis, withdrawal of culprit drug, supportive care, and multidisciplinary management. Currently, there is no gold standard management for SJS/TEN. Direct comparison of the results of clinical studies is difficult due to the lack of uniform study design and measurement standards at different clinical facilities. In addition, the rarity of the disease makes it difficult to perform large-scale studies. Based on the current understanding of the pathophysiology of SJS/TEN, numerous immunosuppressive and immunomodulating treatments have been proposed, including corticosteroids, IVIg, cyclosporine, and TNF-α antagonists.

### 4.4. Systemic Corticosteroids

Corticosteroids are widely used in inflammatory diseases including hypersensitivity. Systemic corticosteroids were one of the first recognized treatments for SJS/TEN. The effectiveness of systemic corticosteroids in the treatment of SJS/TEN has long been debated. Previous studies found that corticosteroids in SJS/TEN patients may increase the risk of infection, overall complications, and mortality [46,47,48]. A European multi-center retrospective study (*n* = 281) found no sufficient evidence of the benefit of corticosteroids [49]. Most meta-analyses have also revealed no beneficial effect of systemic corticosteroid in reducing mortality (Table 1). However, meta-analyses by Zimmermann et al. (2017) and Houschyar et al. (2021) suggested that steroids may improve survival [20,28]. However, Zimmermann et al. showed significant results only based on pooled analysis of individual patient data using an unstratified model.

In a retrospective study of 12 patients with SJS/TEN in the Netherlands by Kardaun et al. (2007), short-term high-dose dexamethasone treatment was found to reduce the mortality rate [50]. Two Japanese studies have also demonstrated the beneficial effect of pulse methylprednisolone therapy with respect to survival and prevention of ocular complications [51,52]. In the study by Hirahara et al., serum levels of IFN-γ, TNF-α, IL-6, and IL-10 in patients with SJS/TEN were decreased after 4 days of methylprednisolone pulse therapy compared with pre-administration levels; however, statistically significant decrease was observed only in IFN-γ and IL-6 levels [52]. In a prospective study conducted in India, 18 patients with TEN were treated with intramuscular injection of dexamethasone (1 mg/kg/day) and all patients survived [53]. In a retrospective study of 85 patients, Mieno et al. found that early introduction (within 4 days from onset) of pulse corticosteroids may reduce severe ocular sequelae [54]. Another retrospective study of 70 SJS/TEN patients also revealed a beneficial effect of corticosteroids regardless of the regimen, i.e., low-dose (≤2 mg/kg/day) or high-dose (>2 mg/kg/day) [55].

Although the beneficial effects of systemic corticosteroids were mostly based on results from retrospective or single-arm non-comparative studies, Japanese treatment guidelines recommend pulse corticosteroid therapy as one of the first-line treatments for SJS/TEN under appropriate infection control [30]. Systemic corticosteroids may be considered life-saving and a low-cost therapy in resource-constrained settings [53].

### 4.5. Intravenous Immunoglobulin (IVIg)

Intravenous immunoglobulin (IVIg), derived from the pooled plasma of healthy donors, is the treatment of choice for both SJS and TEN. IVIg contains a large repertoire of antibody specificities of the donor population. In addition to antibodies with anti-infectious activity, a broad range of naturally occurring autoantibodies in IVIg may regulate important immune functions [56].

Amato et al. first reported the treatment of a SJS patient with IVIg in 1992 [57]. In 1998, Viard et al. demonstrated that IVIg may block Fas-mediated keratinocyte death in vitro, and they also found that IVIg interrupted disease progression and improved prognosis in a cohort of 10 TEN patients [8]. In a subsequent retrospective multi-center study of 12 patients with SJS, IVIg therapy (mean daily dose 0.6 g/kg) for an average of 4 days prevented the progression of epidermal necrolysis and reduced the time to complete mucocutaneous healing [58]. Two other non-randomized single-arm studies have demonstrated the beneficial effect of IVIg [59,60]. However, a large retrospective study of 281 patients in 2008 found no significant difference in mortality on comparing patients treated with IVIg (median total dose: 1.9 g/kg) and those who received supportive care [49]. Several case series and retrospective cohort studies have also reported ineffectiveness of IVIg in reducing mortality or the progression of skin detachment [61,62,63,64].

Generally, total dose greater than 2 g per kg of body weight of IVIg is considered as high-dose regimen for SJS/TEN. In a multi-center retrospective study of 48 patients with TEN, the survival rate of patients treated with high-dose IVIg (mean total dose 2.7 g/kg) was 88% [65]. Trent et al. also reported a significant decrease in mortality rate by 17% with use of high-dose IVIg in a cohort of 16 patients with TEN [66].

Although meta-analysis by Barron et al. found no beneficial effect of IVIg (total dose > 2 g/kg) in decreasing the mortality of SJS/TEN, increasing dose of IVIg was inversely correlated with mortality [17]. In contrast, another two meta-analyses by Huang et al. demonstrated no significant survival benefit of low-dose or high-dose IVIg in patients with TEN [16,19]. The inconsistent results may be attributable to the sensitivity of target cells to Fas, the concentration of IVIg used, and the relative proportions of agonistic and antagonistic anti-Fas autoantibodies in IVIg preparations [67].

### 4.6. Combination of Systemic Corticosteroids and IVIg

In a retrospective study conducted in China, 20 patients receiving a combination of IVIg (0.4 g/kg/day for 5 days) and systemic steroid showed no significant reduction in mortality or the time to taper corticosteroid [68]. In another retrospective study conducted in China, 24 patients treated with high-dose IVIg (>2 g/kg) in combination with systemic steroids showed no significant reduction in standardized mortality ratio compared with those treated with corticosteroids alone [69]. Both these retrospective studies showed no significant benefit of combination therapy in reducing mortality. Conversely, in a prospective open-label study conducted in India enrolling 36 TEN cases, a combination of low-dose IVIg (0.2–0.5 g/kg) and systemic corticosteroids resulted in a significantly lower standardized mortality ratio compared with the corticosteroids-only group [70]. In a multi-center retrospective study, treatment with both steroids and IVIg (mean daily dose 1.0 g/kg) was found to reduce mortality [71]. However, it is difficult to draw any definitive conclusions based on these studies, owing to different study designs and the effect of various confounding factors.

In 2016, Ye et al. conducted a meta-analysis of 26 studies and found that the combination of IVIg and corticosteroid markedly reduced the recovery time but not mortality. Subgroup analysis also revealed a greater effect among Asian patients [18]. In addition, meta-regression analysis by Torres-Navarro et al. showed that IVIg administered in combination with corticosteroids was associated with lesser deaths than predicted by SCORTEN [24]. Two recent network meta-analyses also ranked high priority to combination therapy with IVIg and corticosteroid for reducing mortality among the available systemic therapies [23,26].

### 4.7. Cyclosporine A (CsA)

Cyclosporine A, a calcineurin inhibitor, has drawn attention in recent years. CsA inhibits the activation of CD4+ and CD8+ T cells and subsequently inhibits the release of cytotoxic proteins such as granzyme B, perforin, and granulysin, which play important roles of keratinocyte death in SJS/TEN. In addition, it has anti-apoptotic effects. Therefore, CsA may theoretically benefit patients with SJS/TEN [72].

CsA is less studied compared to systemic corticosteroids for SJS/TEN, and most of the studies were small open-label non-randomized case series. Arevalo et al. reported that treatment with CsA (3 mg/kg/day) was associated with rapid re-epithelialization and a low mortality rate in a case series of 11 TEN patients [73]. In an open-label phase II prospective trial enrolling 29 SJS/TEN patients, no fatality was observed among patients who received CsA (3 mg/kg/day), but some patients required cessation of medication (*n* = 3) or dose-tapering (*n* = 2) due to some adverse effects [74]. The promising effect of CsA was also shown in subsequent studies with an initial dose of 3–5 mg/kg/day with tapering [75,76,77,78,79,80]. However, a large retrospective single-center study of SJS/TEN patients (*n* = 174) found no significant beneficial effect of CsA [81].

Several meta-analyses suggest a beneficial effect of CsA [20,21,22,23,24,27,28]. Some authors also criticized that patients with renal insufficiency, arterial hypertension, uncontrolled diabetes mellitus, severe infection, malignancy, or immunodeficiency were excluded from some series, and this may have introduced an element of selection bias. Moreover, the results of meta-analysis are likely to have been affected by publication bias [82]. There are also concerns pertaining to the renal and hepatic toxicity of CsA. CsA should be avoided or used with caution in patients with pre-existing renal insufficiency (creatinine clearance < 60 mL/min) or uncontrolled diabetes [72].

### 4.8. TNF-Alpha (TNF-α) Inhibitors

TNF-α is not only involved in upregulation of FasL in keratinocytes but also acts as a death receptor ligand by itself. Besides, TNFα was shown to enhance HLA class I expression on keratinocytes, rendering them more susceptible to T cell-mediated cytotoxicity [83]. The increased TNF-α levels in SJS/TEN patients has led to the suggestions of using TNF-α inhibitors [84]. However, a well-known randomized controlled trial which used thalidomide for SJS/TEN treatment was terminated early due to excessive deaths in the treatment arm. Thalidomide is a potent inhibitor of TNF-α in vitro and in vivo, but paradoxical enhancement of TNF-α production was observed in the group treated with thalidomide [85].

Some case reports and case series have described beneficial effects of biologic TNF-α inhibitors such as etanercept (a soluble fusion protein) and infliximab (anti-TNF-α monoclonal antibody) [86,87,88,89,90,91,92,93,94,95]. In a randomized controlled trial enrolling 96 patients with SJS/TEN, patients treated with etanercept showed significantly lower mortality rate. Compared to corticosteroids, etanercept reduced the skin-healing time and the incidence of gastrointestinal hemorrhage. Moreover, etanercept significantly decreased the levels of TNF-α and granulysin in blister fluids and plasma and increased the regulatory T cell population in the peripheral blood [96].

Several meta-analyses have also found that etanercept and infliximab reduced mortality [22,23,27]. A systematic review by Sachdeva et al. [25] also recommended monotherapy with TNF-α inhibitors, especially etanercept, based on the results of the above-mentioned trial [96]. However, some other meta-analyses did not find beneficial effects of biologic anti-TNF-α for SJS/TEN [24,26,28], and the major concern was that most of the published studies were case reports and case series. More robust studies are required to confirm the efficacy of these drugs in the treatment of SJS/TEN.

### 4.9. Combination of Biologic Anti-TNF-α and Corticosteroids

Few case reports have described the use of combination therapy with anti-TNF-α and corticosteroids [95,96,97,98,99,100,101,102,103]. In 2021, Sachdeva conducted a systematic review of 38 studies (including 27 case reports) and found that both biologic monotherapy and combination therapy were associated with improved outcomes in SJS/TEN [25]. In 2022, Ao et al. recruited 25 patients with SJS/TEN and found that combination therapy with etanercept and corticosteroids significantly shortened the duration of acute phase, hospital stay, and skin re-epithelialization in comparison to a corticosteroid monotherapy group. Both treatments significantly reduced the serum levels of IL-15, but the combination therapy also decreased the serum levels of IL-6 and IL-18 [104]. Zhang et al. retrospectively enrolled 242 SJS/TEN patients from Taiwan and China and found that patients who received combination therapy with etanercept and corticosteroids had lower mortality rates in comparison with corticosteroid alone or IVIg in combination with corticosteroids [105].

### 4.10. Combination of Biologic Anti-TNF-α with Other Treatments

A few case reports have described the use of combination therapy with anti-TNF-α and IVIg [90,106,107]. Pham et al. reported that addition of etanercept to IVIg plus supportive care may improve outcomes compared to IVIg with supportive care alone in a case series of 13 patients [108].

Several case reports have also described the use of a combination of anti-TNF-α and systemic treatments. López-Gómez et al. reported using combination therapy with etanercept and cyclosporine in a ribociclib-related SJS patient [109]. Gavigan et al. and Coulombe et al. reported using combination therapy with etanercept, cyclosporine, and corticosteroids [110,111]. Sibbald et al. and Holtz et al. reported using combination therapy with etanercept, IVIg and corticosteroids [112,113] In 2014, Paquet et al. conducted a randomized controlled study comparing the efficacy of N-acetylcysteine alone and combination therapy with N-acetylcysteine and infliximab in patients with TEN (*n* = 5 in each group). Although no drug-induced adverse effects were observed in the combination group, no reversal of disease progression was found [114].

### 4.11. Plasmapheresis

The mechanism of plasmapheresis is the removal of drug, drug metabolites, and cytokines from the patient. The Japanese guidelines recommend systematic steroids, IVIg, and plasmapheresis as the three first-line treatments of choice [30]. Plasmapheresis is a safe treatment and can be performed daily or every other day with few adverse side effects. Some case reports or series have reported the efficacy of plasmapheresis in SJS/TEN treatment [115,116,117,118,119,120]. Narita et al. reported a decrease in the serum levels of pro-inflammatory cytokines after plasmapheresis [121]. Han et al. conducted a prospective observational study of 28 patients with TEN or SJS/TEN overlap; they reported that plasmapheresis was superior to conventional therapies (such as IVIg or corticosteroids) with respect to reducing the mortality and the duration of hospital stay [122]. However, an observational study by Furubacke et al. (*n* = 8) found no benefits in terms of mortality, length of stay, or time to re-epithelialization [123].

In 2017, Giudice et al. reported the safety and efficacy of the combination of plasmapheresis and CsA in the management of TEN at the burn unit of the University of Bari with 12 TEN patients enrolled [124]. Krajewski et al. and Lissia et al. reported that the combination of plasmapheresis with IVIg may improve outcomes [125,126]. A network meta-analysis by Tsai et al. indicated that IVIg combined with plasmapheresis is a potentially effective option but more evidence is required to draw definitive conclusions [26].

## 5. Limitations

Based on the current literature, there is at least some evidence to support the use of immunomodulating agents in the treatment of SJS/TEN. Absence of high-quality data, underlying diseases, lack of treatment access, financial constraints, and experience of prescribers may influence the choice of treatments. For example, etanercept is easily available and costs less than IVIg. We suggested a simple algorithm for choosing systemic treatments (Figure 1) and this algorithm could be modified with emergence of new evidence of efficacy.

There are significant limitations to the use of evidence from observational studies, which may be prone to bias. Heterogenicity among the studies with respect to disease phase, difference in dressing regimens used, treatment setting (burn unit or intensive care unit), and underlying diseases may all affect the results. There may also be publication bias. Ethical constraints in study design such as assignment of high-risk patients to control groups may also be a problem. Previous studies have not considered these points, which may have led to discrepant results. Some meta-analysis studies used SCORTEN (published in 2000)-predicted death as comparison. However, with the improvement of supportive care, the predicted mortality rate may be less in the supportive care group in recent years. Establishing expert consensus or guidelines and registry systems with multi-center prospective study may help clarify the effectiveness of treatment.

## 6. Conclusions

This review summarizes recent advances in the pathophysiology, diagnosis, and treatment of SJS/TEN. SJS/TEN is a severe adverse drug reaction associated with a high mortality rate and its treatment algorithm has not been well established.

The use of systemic corticosteroids and IVIg is still contested. However, an increasing number of recent studies have suggested the effectiveness of cyclosporine or biologic anti-TNF-α. Accumulation of more data of these treatments is desirable. Finally, the pathogenesis of SJS/TEN has been elucidated in recent years and the breakthrough of these studies may help identify promising targets for the discovery of novel therapeutic agents.

## Figures and Tables

**Figure 1 biomedicines-10-02105-f001:**
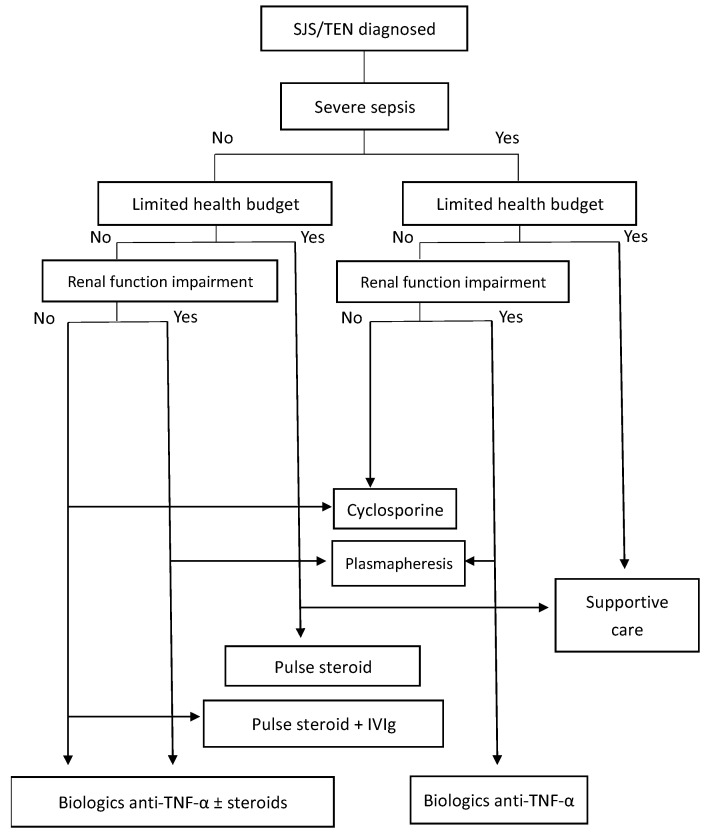
Algorithm for choosing systemic treatments for SJS/TEN.

**Table 1 biomedicines-10-02105-t001:** Summary of published systematic review and meta-analysis.

Authors	Included Studies/Treatment	Summary of Potential Treatments with Benefits
Huang et al., 2012 [16]	17 studies/IVIg	Both high-dose and low-dose IVIg were not associated with survival benefit.
Barron et al., 2015 [17]	13 studies/IVIg	Increasing dose of IVIg was associated with decreased mortality.
Ye et al., 2016 [18]	26 studies/IVIg + corticosteroid	Combination of IVIg and corticosteroid markedly reduced recovery time but not mortality.
Huang et al., 2016 [19]	11 studies/IVIg	IVIg was ineffective in reducing mortality in TEN patients, even at high-dose.
Zimmermann et al., 2017 [20]	96 studies/multiple	Glucocorticoids and cyclosporine were the most promising treatment.
Ng et al., 2018 [21]	9 studies/cyclosporine	Cyclosporine significantly reduced mortality.
Zhang et al., 2019 [22]	27 studies/TNF-α inhibitors	Biologic TNF-α inhibitors (infliximab and etanercept) are safe and effective treatments.
Patel et al., 2021 [23]	24 studies/multiple	Cyclosporine reduced mortality in TEN patients. Etanercept and combination of IVIg and corticosteroid and were also promising.
Torres-Navarro et al., 2021 [24]	38 studies/multiple	The meta-regression analysis confirmed that cyclosporine and combination of IVIg and corticosteroid were associated with less deaths than predicted by SCORTEN.
Sachdeva et al., 2021 [25]	38 studies/biologics	TNF-α inhibitors monotherapy improved outcomes and may be safer compared to combination therapy.
Tsai et al., 2021 [26]	66 studies/multiple	Combination of IVIg and corticosteroid was the only treatment with significant survival benefits.
Krajewski et al., 2022 [27]	42 studies/multiple	The lowest mortality was found in etanercept group followed by cyclosporine.
Houschyar et al., 2021 [28]	16 studies/multiple	Systemic glucocorticoids showed a survival benefit. Cyclosporine also showed promising results.

IVIg, intravenous immunoglobulin; SCORTEN, severity-of-illness score for toxic epidermal necrolysis; TEN, toxic epidermal necrolysis; TNF-α, tumor necrosis factor alpha.

## Data Availability

All data were collected from published articles available in the public domain.

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
