# Peer review of "A Review of the Systemic Treatment of Stevens–Johnson Syndrome and Toxic Epidermal Necrolysis"

_biomedicines, 2022, doi:10.3390/biomedicines10092105_

Round 1

Reviewer 1 Report

In the paper, authors described pathogenesis, management and treatment of SJS/TEN, attempting to summarize existing evidence-based data on systemic therapies. This is important topic as SJS/TEN is life-threating condition, but also very challenging as SJS/TEN is extremely rare. Therefore conducting a well-organized clinical trial is very difficult, so studies in this field are often of poor methodological quality, which was described in the article.

I have the following remarks, which might improve value of this paper for readers:

- For many times authors stated that results of clinical trials on particular treatment options are conflicting. The authors mentioned shortages of papers and sources of biases (eg. lines 268-270). But it would be very welcome to interpret more in detail these findings from literature, also to justify the authors’ proposal for algorithm (Fig. 1). In other words, basing on authors expertise and experience: which of these many studies/ papers are the most important, the most reliable for making clinical decisions, taking into account for instance number of patients included, quality of study design (eg. case-with reliable control), reported inclusion criteria, confounding factors, or clinical significance of end-points under evaluation (mortality rate vs cytokine levels)

Minor issues:

- line 356 – typo

- line 357 – typo? sue/use

- Tab. 1. – references would be helpful for readers (except from an author of study and a year of publication);

- Tab. 1 – “biologics” would be helpful to know what biologics (table should be self-explanatory)

- About a half of the paper is dedicated to very general info on SJS/TEN (lines 1-209), the second half to systemic treatment. So, the content of the paper does not correspond precisely to the title (“A review on systemic treatment …”)

Reviewer 2 Report

It's a very interesting paper with focus on systemic treatment. Nobody has a real answer and so we need these kind of study.

You compare the studies well and we can learn how to treat better our patients. We need more studies.

I found only one mistake on line 222: you wrote Sylvia but this is her name, the surname is Kardaun.

Reviewer 3 Report

Chang et al. present a review in the systemic treatment of Stevens Johnosn syndrome and toxic epidermal necrolysis.

Following comments:

Please explain the term „study“ as both RCT and case studies are included.

Please explain (or speculate on) differences between Asia and Europe with respect to therapeutic responses.

How relevant is the time of intervention that is how early therapy is initiated? How is this aspect regarded in the pertinent publications?
line 53: which are cutaneous „problems“? Those mentioned in line 55?

Please mention and summarize available guidelines (those from Japan and UK are mentioned).

Line 129: the chapter should be divided as the major part refers to diagnostics, the second part to SCORTEN, the third part to supportive therapy.

Please refer to the choice of intensive care units or burn units. Which to prefer at what time point and based on which criteria?
line 311: better write „arterial hypertension“.

Line 316; what is CrCL? Only used once, so use full wording, not abbreviation.

Line 336: which TNFB to use? Please discuss.

Line 356 ff: which were the clinical effects?

Line 388: how many studies have been included?

Table 1: what are the respective citations?
Figure 1: severe sepsis, limited budget just supportive therapy??

Round 2

Reviewer 3 Report

all major comments have been followed